# Bone Marrow Aspirate Concentrate versus Platelet Rich Plasma or Hyaluronic Acid for the Treatment of Knee Osteoarthritis

**DOI:** 10.3390/medicina57111193

**Published:** 2021-11-02

**Authors:** Oliver Dulic, Predrag Rasovic, Ivica Lalic, Vaso Kecojevic, Gordan Gavrilovic, Dzihan Abazovic, Dusan Maric, Mladen Miskulin, Marko Bumbasirevic

**Affiliations:** 1Medical Faculty, University of Belgrade, 11000 Belgrade, Serbia; marko.bumbasirevic@gmail.com; 2Department for Orthopedic Surgery and Traumatology, Clinical Center of Vojvodina, Hajduk Veljkova 1-9, 21000 Novi Sad, Serbia; rasovicpedja@gmail.com (P.R.); laleort021@gmail.com (I.L.); keckons@gmail.com (V.K.); 3Medical Faculty, University of Novi Sad, 21000 Novi Sad, Serbia; ducamaric@gmail.com; 4Atlas Hospital, 11000 Belgrade, Serbia; ggavra45@gmail.com; 5MD Orto Hospital, 21000 Novi Sad, Serbia; 6Renova Hospital Belgrade, 11000 Belgrade, Serbia; adzihan@gmail.com; 7Specijalna Bolnica za Neurokirurgiju i Ortopediju Aksis, 10000 Zagreb, Croatia; mladen.miskulin@aksis.hr; 8Medical Faculty, University of Split, 21000 Split, Croatia; 9Clinic for Orthopedic Surgery and Traumatology, Clinical Center of Serbia, 11000 Belgrade, Serbia

**Keywords:** bone marrow aspirate concentrate, platelet rich plasma, hyaluronic acid, knee osteoarthritis, regenerative medicine, stem cells

## Abstract

*Background:* In the last decade, regenerative therapies have become one of the leading disease modifying options for treatment of knee osteoarthritis (OA). Still, there is a lack of trials with a direct comparison of different biological treatments. Our aim was to directly compare clinical outcomes of knee injections of Bone Marrow Aspirate Concentrate (BMAC), Platelet-rich Plasma (PRP), or Hyaluronic acid (HA) in the OA treatment. *Methods:* Patients with knee pain and osteoarthritis KL grade II to IV were randomized to receive a BMAC, PRP, and HA injection in the knee. VAS, WOMAC, KOOS, and IKDC scores were used to establish baseline values at 1, 3, 6, 9, and 12 months. All side effects were reported. *Results:* A total of 175 patients with a knee osteoarthritis KL grade II-IV were randomized; 111 were treated with BMAC injection, 30 with HA injection, and 34 patients with PRP injection. There were no differences between these groups when considering KL grade, BMI, age, or gender. There were no serious side effects. The mean VAS scores after 3, 7, 14, and 21 days showed significant differences between groups with a drop of VAS in all groups but with a difference in the BMAC group in comparison to other groups (*p* < 0.001). There were high statistically significant differences between baseline scores and those after 12 months (*p* < 0.001) in WOMAC, KOOS, KOOS pain, and IKDC scores, and in addition, there were differences between these scores in the BMAC group in comparison with other groups, except for the PRP group in WOMAC and the partial IKDC score. There were no differences between the HA and PRP groups, although PRP showed a higher level of clinical improvement. *Conclusions**:* Bone marrow aspirate concentrate, Leukocyte rich Platelet Rich Plasma, and Hyaluronic acid injections are safe therapeutic options for knee OA and provide positive clinical outcomes after 12 months in comparison with findings preceding the intervention. BMAC could be better in terms of clinical improvements in the treatment of knee OA than PRP and HA up to 12 months. PRP provides better outcomes than HA during the observation period, but these results are not statistically significant. More randomized controlled trials and high quality comparative studies are needed for direct correlative conclusions.

## 1. Introduction

A therapeutic option for osteoarthritis (OA) of the knee comprises pain management, physical therapy with life-modifying recommendations, joint injections, and joint replacement for the end-stage phase. Intraarticular injections have several advantages over systemic delivery, including increased local bioavailability, reduced systemic exposure, fewer adverse events, and reduced cost [1]. Three injectable materials have been widely used for intra-articular treatment of the knee OA: corticosteroids (with or without local anesthetics), hyaluronic acid (HA) preparations, and in the last decade, bioregenerative preparations, such as TNF and Il-1 inhibitors, platelet-rich plasma injections, bone marrow-derived stem cells, adipose-derived stem cells, and amnion-derived mesenchymal stem cells [1,2].

Bone marrow mesenchymal stem cells are a heterogeneous mixture of cells with at least two different functions. Some of these cells are already involved in the osteogenic pathway and accelerate bone formation and regenerative repair [3,4,5], whereas other MSCs have the capacity of acting as immunomodulatory and trophic factors [6]. Bone marrow concentrate contains very few mesenchymal stem cells (only 0.001–0.01% of the cellular content of BMAC are stromal cells) [7], predominantly hematopoietic stem cells, platelets (with its growth factors), and cytokines. Bone marrow cells consist of erythroblasts, neutrophils, eosinophils, basophils, monoid cells (monocytes containing mesenchymal stem cells and macrophages), lymphocytes, and plasma cells [8]. These cells are present in various stages of differentiation [9]. The hematopoietic progenitor cells can morph into mesenchymal stem cells, differentiate into chondrocytes, and are more osteo-inductive than adipose-derived cells [10]. Following a knowledge base derived from pre-clinical, basic studies about the role of MSC in an inflammatory environment in the knee, a rationale for the treatment of OA with stem cells from different sources is justified [11,12,13]. PRP is defined as the portion of the plasma fraction of blood having a platelet concentration above the baseline value [14]. Platelet-rich plasma is an autologous derivative of whole blood that contains very high concentrations of growth factors such as transforming growth factor-b, insulin-like growth factor, fibroblast growth factor, platelet-derived growth factor, and vascular endothelial growth factor, as well as bioactive proteins that influence the healing of tendon, ligament, muscle, and bone [15]. The current consensus is based on a simple classification system dividing the many products in four main families, based on their fibrin architecture and cell content: Pure Platelet-Rich Plasma (P-PRP), Leukocyte- and Platelet-Rich Plasma (L-PRP), Pure Platelet-Rich Fibrin (P-PRF), and Leukocyte- and Platelet-Rich Fibrin (L-PRF) [16]. On the basis of the current evidence, PRP injections reduced pain more effectively than did placebo injections in OA of the knee [17,18]. HA is a natural carbohydrate found in the human body, with a molecular weight of 846,786 g/mol. HA is an amorphous, glassy substance and is part of the class of glycosaminoglycans or acid mucopolysaccharide compounds, with the effect of filling the extracellular spaces between the collagen fibers. HA behavior in biological structures is to attract water, lubricate intracellular structures, and give “volume,” forming a gelatinous matrix with which the elastin and collagen fibers are coagulated and aligned together. IA-HA has been proposed to have many therapeutic mechanisms of action in the OA knee, including shock absorption, joint lubrication, anti-inflammatory effects, chondroprotection, proteoglycan synthesis, and cartilage matrix alterations [19,20,21,22].

The aim of this study was to evaluate clinical effects of BMAC, PRP, and HA therapy on knee OA and to compare clinical results of these regenerative treatment options.

Null Hypothesis: Having in mind that there was only one direct comparison of BMAC and PRP in literature, our hypothesis was that BMAC could be a more effective therapy because of a higher number of different cells with an active biological role and a higher level of bioactive protein molecules.

## 2. Materials and Methods

### 2.1. Study Design and Population

This was a single-center, prospective clinical trial with 3 study arms. From April 2016 to December 2017, outpatients with a history of complaints of knee pain were subjected to a thorough clinical history, physical examination, laboratory test, and X-ray. The study was performed solely in cases when the patients understood and agreed to the treatment method and procedure by signing an informed consent. After careful review of all the test results, we set inclusion criteria for this study: (1) adult patients with symptomatic knee OA, Kellgren-Lawrence (KL) grade from 2 to 4 with (3) symptoms for at least 12 months. Exclusion criteria was also set in this study: (1) knee instability, (2) severe misalignment, (3) inflammatory arthritis such as rheumatoid arthritis and ankylosing spondylitis, and (4) underlying diseases such as hematologic disorders, septicemia, coagulopathy, neoplasm, active infection, and immune deficiency. According to these criteria, 195 patients who were diagnosed with OA were included in this study and were sequentially selected and treated. During a study process from allocation to the final follow-up after twelve months, 20 patients were lost for follow-up and were not included in the final calculations. In the first round, during the 2016 period, patients were treated with BMAC and in the second round, in 2017, when we were supplied with PRP sets and HA injections, patients were randomly allocated in two groups: HA and PRP groups. For randomization at this stage, we used R.3.3.0 version software. All groups were monitored using the same methodological criteria. The study was performed according to the Declaration of Helsinki for medical research involving human subjects. An approval by the Research Ethics Committee has been provided (EC KCV 319-321/16). The study was registered in ClinicalTrials.gov (NCT03825133) and the results presented in this paper are an excerpt from this registered study.

### 2.2. BMAC Processing Procedure

For the procedure, the patient was placed in supine position, following preparation and draping of tibial tubercle, and then local anesthesia (Lidocaine 2 mL + Marcaine 10 mL) from skin to periosteum was infiltrated. A small incision (11 blade) was performed and trocar was placed in the cancellous part of the bone. Two 50 mL syringes were prepared with 10 mL of anticoagulant, acid citrate dextrose formula A (ACD-a). Moving trocar in the bottom-up direction, in order to provide more harvesting sites, about 80 mL of autologous bone marrow was aspirated (two syringes of bone marrow; in total, 40 mL + 10 mL of anticoagulant). Bone marrow was filtrated through a 150 micron filter in order to remove coagulum and potential bone fragments. The filtrated bone marrow was processed using the Arthrex Angel separating system, following which, an end product of about 5–6 mL of BMAC was obtained. Moving further, an injection of BMAC was injected into the treated knee.

### 2.3. In-Vitro Experiments

Quality control analysis for every single sample was performed. The assessment of the total nucleated cells (TNC) count was performed and analysis of cell viability was done using 7-Amino Actinomycin D (7-AAD) fluorochrome for all samples. For 20 randomly selected samples, flow cytometry immunophenotyping was conducted using antihuman mAbs in different combinations for multicolor analysis of the samples CD3, CD4, CD8, CD19, CD56, CD14, CD10, CD45, CD34, CD90, CD73, CD133, CD105, and CD271. The flow cytometry was done using a Beckman Coulter FC 500 flow cytometer with CXP analysis software.

Multiparameter flowcytometry quality control analysis of BMAC samples showed a very high level of viability (98.2 ± 0.7% of cells showed no staining for 7AAD dye) with TNC in BMAC samples 25 ± 6.32 × 10^6^/mL. Analysis of specific CD34 + CD45 + HSC and CD34-CD45-MSC populations markers in randomly selected 20 BMAC samples showed very high level of viability (98.2 ± 0.7% of cells showed no staining for 7AAD dye) with TNC in BMAC samples 25 ± 6.32 × 10^6^/mL. Analysis of specific CD34 + CD45 + HSC and CD34-CD45-MSC populations markers in randomly selected 20 BMAC samples showed presence of CD271 + CD90 + HSC and CD271 + CD90 + MSC cells (20,830.33 ± 34,892.29 cells/mL BMAC, 2775.045 ± 3920.336 cells/mL BMAC, respectively).

### 2.4. PRP Procedure

For the preparation of PRP, 60 mL of peripheral blood was taken into syringes containing acidum citrate dextrose formula A (ACD-A) in ratio 7:1. The whole blood was processed using a fully automated system Angel^®^ (Arthrex^®^) with G force and time of spinning automatically adjusted in order to sequester desired volume of blood into three fractions, PRP, RBC (Red Blood Cells), and PPP (Platelet Poor Plasma), and to obtain PRP substrate with platelet number 6–8 folds higher than baseline, which was automatically set on the device. An average platelet count rate in the PRP group was 320.37 × 10^6^ and the leukocyte (Le) rate was 7.07 × 10^6^. Following the centrifugation and concentration procedure, an average rate of platelets was 2179.31 × 10^6^ and the average rate of leukocytes was 16.16 × 10^6^. Average concentration rate for platelets was 7.23× baseline and for Le it was 2.22× baseline, so our formulation can be defined as Leukocyte Rich Platelet rich plasma (LR-PRP).

### 2.5. HA Injection

For this group, HA with the Cartinorm^®^ brand name was injected. Cartinorm^®^ is manufactured by Goodwill Pharma, Hungary. It is a viscous solution consisting of a high molecular weight fraction of purified 1% natural sodium hyaluronate (4000 kDa) in buffered physiological sodium chloride, having a pH of 6.8–7.6. The sodium hyaluronate was supplied as a sterile, non-pyogenic solution in 2 mL pre-filled syringes containing 20 mg of sodium hyaluronate, 16.6 mg of sodium chloride, 0.52 mg of potassium dihydrogen phosphate, 2.8 mg of dibasic sodium phosphate dihydrate, and up to 2 cc water for injection. Injections were administered to a total of 3 treatments, 1 every week for 3 weeks.

### 2.6. Data Collection

Patients were asked to self-report comfort with the procedure (VAS sting pain) and to report pain before intervention and after 3, 7, 14, and 21 days using VAS (Visual Analog Scale), and they were scheduled to be followed-up after 1, 3, 6, 9, and 12 months. In the HA group, VAS was measured after the third injection. On follow-ups, clinical results were gathered using the WOMAC (Western Ontario and McMaster Universities Osteoarthritis Index, version 3.1), KOOS (Knee injury and Osteoarthritis Outcome Score), and IKDC (International Knee Documentation Committee) score. Twelve months later, patients were asked if they would repeat the intervention by having the opportunity to choose from three possible answers: yes, no, or maybe. Patients were asked to report any potential adverse events starting from the procedure to the end of the study. Adverse events were described as conditions connected to the procedure itself or to the nature of the disease.

### 2.7. Data Analysis

Analysis of primary data was done by descriptive statistical methods and methods for testing statistical hypothesis. Results are presented as frequency (percent), median (range), and mean ± SD. We applied: chi-square test, one-way ANOVA, and repeated-measures ANOVA as methods for testing the statistical hypothesis. Statistical hypotheses were analyzed at the level of significance of 0.05. Statistical data analysis was performed using IBM SPSS Statistics 22 (IBM Corporation, Armonk, NY, USA).

## 3. Results

A total of 263 patients were assessed for eligibility and 195 were allocated for a procedure. A total number of 175 patients were treated and screened for period of 12 months (see flow diagram—Figure 1) and fulfilled pre-intervention tests. For the BMAC, HA, and PRP groups, a total number of 111, 30, and 34 patients, respectively, fulfilled the post-intervention knee VAS pain scale and completed all aspects of the procedure from baseline data to 12 months follow up. In the BMAC, HA, and PRP groups, 11, 2, and 4 patients, respectively, reported slight swelling up to 7 days after intervention without any consequences. There were no other serious adverse events. Patients’ characteristics are shown in Table 1.

The overall results for knee VAS pain are presented in Table 2. Using the Bonferroni test intergroup comparison, there was no significance in VAS pain in the three groups before intervention, but there was significance between the BMAC group and the PRP (Mean Difference (MD) = −2.523, *p* < 0.001) and HA group (MD = −2.248, *p* < 0.001) after 3 days. After 7 days, between the BMAC and PRP groups, there was a difference (MD = −3.010, *p* < 0.001), and the same as between the BMAC and HA groups (MD = −2.763, *p* < 0.001); same findings we discovered after 14 and 21 days (*p* < 0.001). These results are visualized in Figure 2. There is a clear significance if we compare pre intervention values and values after 3, 7 14, and 21 days for the BMAC, Ha, and PRP groups, respectively (*p* < 0.001).

All results for clinical tests are presented in Table 3. Regarding KOOS pain score, inter-group comparison using the Bonferroni post-hoc test demonstrated significance in pre intervention measurement among the BMAC and HA groups (BMAC group had higher value, 9.154; *p* = 0.036), but there was no difference between the BMAC and PRP groups (MD = 6.284; *p* = 0.207). Distribution of values for all three interventions is shown in Figure 3. After intervention, there was high significance between the BMAC group and both the HA and PRP groups after one month (BMAC vs. HA = MD = 18.214; *p* < 0.001; BMAC vs. PRP = 10.829; *p* = 0.012), after 3 months (BMAC vs. HA = 12.184; *p* = 0.006; BMAC vs. PRP = 12.364; *p* = 0.008), and after 6 months (BMAC vs. HA = 17.222; *p* < 0.001; BMAC vs. PRP = 13.009; *p* = 0.004). After this period, there was significance between the BMAC and HA groups after 9 months (BMAC vs. HA = 17.413; *p* < 0.001), but no difference in the measurement of BMAC vs. PRP (MD = 8.705; *p* = 0.101). At 12 months, there were differences between BMAC and HA (BMAC vs. HA =15.301; *p* = 0.002), but not in comparison to BMAC vs. PRP = 8.191, *p* = 0.170; although, the BMAC group showed better results. There was no significance between the HA and PRP groups in all measurements. On the other hand, there was high significance in comparison with pre-intervention values for all three therapies and up to 12 months (at 12 months for BMAC MD = −24.380; *p* < 0.001; for HA MD = −18.233; *p* < 0.001; and for PRP MD = −22.474; *p* < 0.001). There were no significance in values for all three therapies comparing values at 1 month follow up and other later measurements.

Regarding the KOOS overall score, inter-group comparison using the Bonferroni post-hoc test found no significance in pre-intervention measurement among groups. After intervention, there was high significance between BMAC and both the HA and PRP groups (at one month BMAC vs. HA = 15.516; *p* < 0.001; BMAC vs. PRP = 9.087; *p* = 0.035; at 3 months BMAC vs. HA = 20.237; *p* < 0.001; BMAC vs. PRP = 12.404; *p* = 0.004; and after 6 months—BMAC vs. HA = 19.298; *p* < 0.001; BMAC vs. PRP without significance = 9.151; *p* = 0.054). After this period, there was significance between the BMAC and HA groups to the end of the observation period, and no significance between the BMAC and PRP groups, although the BMAC group showed better results. There was no significance between the HA and PRP groups in all measurements (Figure 4). Again, there was significance in pre-intervention values of BMAC, HA, and PRP therapies comparing at 12 months (after 12 months comparing with pre-intervention values: BMAC = −25.345; *p* < 0.001; HA = −19.456; *p* = 0.001; and PRP = −15.586; *p* = 0.010). Interestingly, after one and three months, in group PRP, there was an increase in KOOS overall score, but this increase was not statistically different in comparison with pre intervention values. After 3 months, there was high significance comparing pre-interventional measures. There were no significance in values for all three therapies comparing values on 1 month follow up and other later measurements.

Regarding the WOMAC score, inter-group comparison using the Bonferroni post-hoc test found no significance in pre-intervention measurement among groups. After intervention, there were high significance between BMAC and HA until 12 months, but no significance between the BMAC and PRP groups up to 12 months, although the BMAC group showed better results (at one month BMAC vs. HA = −12.085; *p* < 0.007; BMAC vs. PRP = −6.718; *p* = 0.214; at 3 months BMAC vs. HA= −12.415; *p* = 0.011; BMAC vs. PRP = −7.199; *p* = 0.227; at 6 months: BMAC vs. HA= −12.504; *p* = 0.008; BMAC vs. PRP without significance = −6.000; *p* = 0.384; at 9 months: BMAC vs. HA = −12.735; *p* = 0.009; BMAC vs. PRP without significance = −6.979; *p* = 0.255; and at 12 months: BMAC vs. HA-MD = 10.949; *p* = 0.035; BMAC vs. PRP without significance= −6.720; *p* = 0.306). Once, there was no significance between the HA and PRP groups in all measurements (Figure 5). If we compare pre-intervention values and at 12 months, we found high significance in pre-intervention values of BMAC and PRP therapies and up to 12 months (for BMAC: MD = 20.007; *p* < 0.001; for PRP: MD: 17.064; *p* = 0.001). Interestingly, in the HA therapy group, there was a drop of WOMAC when comparing before and after intervention, but significance was observed only at 12 months follow up (MD: 11.125; *p* = 0.048). There was no significance in values for all three therapies comparing values at 1 month follow up and other later measurements except in the HA group in comparison between values on 9 and 12 months.

Regarding the IKDC score, inter-group comparison using the Bonferroni post-hoc test demonstrated significance in pre-intervention measurement among the BMAC and HA groups (BMAC group had higher value, MD = 7.622; *p* = 0.020). After intervention, there was high significance between BMAC and both the HA and PRP groups (at one month: BMAC vs. HA = −13.046; *p* = 0.001; BMAC vs. PRP without differences = 6.989; *p* = 0.115; at 3 months BMAC vs. HA = 17.892; *p* < 0.001; BMAC vs. PRP = 12.354; *p* = 0.002; at 6 months: BMAC vs. HA = 20.794; *p* < 0.001; BMAC vs. PRP = 9.728; *p* = 0.033; at 9 months: BMAC vs. HA = 18.716; *p* < 0.001; BMAC vs. PRP without significance = 8.452; *p* = 0.089; and at 12 months: BMAC vs. HA = 15.224; *p* = 0.002; BMAC vs. PRP without significance = 9.070; *p* = 0.086). Even though there was no permanent significance between the BMAC and PRP groups at all measures, the BMAC group showed better results. There was no any significance between the HA and PRP groups in all measurements, but PRP showed somewhat better results (Figure 6). When we compared pre-intervention values with those at 12 months, we found that they showed high significance in pre-intervention values for the BMAC and PRP therapies groups at 12 months (BMAC: MD = −21.426; *p* < 0.001; PRP: MD = −15.659; *p* < 0.001). Again, in the HA therapy group, there was an increase of IKDC values when comparing before and after intervention, but significance was observed only at 12 months follow up (MD= −13.824; *p* = 0.003). There was no significance in values for all three therapies comparing values at 1 month follow-up and other later measurements except again in the HA group in comparison between values at 9 and 12 months.

## 4. Discussion

This study revealed clinical changes after the treatment of knee OA with BMAC, PRP, and HA, showing improvement in clinical findings in comparison with pre-treatment values at 12 months. Furthermore, this study showed that BMAC treatment could be more effective in comparison with the other two biological options starting from 3 days after therapy up to 12 months.

To our knowledge, this is the second direct comparison of clinical findings regarding these three popular bioregenerative treatments. In the recently published study of Anz et al. [23], methodologically very similar to our study, they concluded that both LR-PRP and BMC were effective in improving patient-reported outcomes in patients with mild to moderate knee OA for at least 12 months, but neither treatment provided a superior clinical benefit. In this study, significant effect of time was observed on all WOMAC subscales and IKDC. Both groups significantly improved on all WOMAC subscales and IKDC from baseline to 1-month follow-up, with no further significant improvement in scores. When we compared our results with these results, we found the same findings in the WOMAC scale (no differences between BMAC and PRP in all time frames) and IKDC (no differences at 9 and 12 months measurements), although BMAC showed better results and showed differences in the KOOS and VAS scales. Slight divergences in the mentioned study comparing with our results could be explained with a fact that, in Anz et al.’s study, both group of patients had a lower KL grade (the total KL grade was 1.8 +/− 0.7, in BMA group was 1.8, and in L-PRP group 1, 9), which is significantly different compared to our average KL grade values (BMAC = 3, PRP = 2.82).

We found almost immediate pain relief after BMAC injections, with a trend of pain decrease over measured time (from 3 days after to 3 weeks). A drop in pain level was found among PRP and HA groups, but it was not as sharp as we found in the BMAC group. Tissue damage and inflammation, typical for OA, result in release of a wide array of mediators that can bind specific receptors on nociceptors that innervate the affected tissues, resulting in different biological effects, including: neuronal excitation, eliciting pain; peripheral sensitization; and release of neuropeptides such as substance P and calcitonin-gene related peptide (CGRP, which contributes to neurogenic inflammation).

Excessive neuronal activity of primary sensory neurons can also trigger neuroinflammation, which is characterized by the activation of satellite glial cells and infiltration by immune cells such as macrophages in the dorsal root ganglia (DRG), where the cell bodies of the sensory neurons reside. In OA, many potential pain relief mechanisms have been described as a blockade of nerve growth factor (NGF) activity, changes in the property of ion channels, activation of the G-protein coupled receptors (GPCR), and many other targets [24,25]. Although there are no well described mechanisms of the action of cells from BMAC (or their excreted products) and other bioactive substances, we believe that such an immediate pain relief after BMAC injection is probably related to an action on some pain related pathways. 

There were no adverse events among our patients related to either of the three procedures during a period of 12 months. Safety of the BMAC therapy was investigated in many trials, both for scientific, clinical, or regulatory purposes [11,26,27]. Our safety results are like previously published reports of BMAC use in osteoarthritic knees. There were no clinical evidences to suggest that treatment with MSCs (alone or in mixture with other stem cells or PRP) increase the risk of neoplasm, immunological, or other related diseases. Centeno C. et al. [26], performing an investigation in multi-center analysis among 2372 adults undergoing autologous stem cell therapy, concluded that the rate of reported neoplasms is even lower in the treatment group in comparison with the general population [26].

Lack of any adverse events is expected; BMAC, with its ingredients, is fully autologous, so there is not one substance that could elicit foreign body reactions beside rare cases [28], immunological attacks, or toxic spreads. With these interventions, we simply transplant specific body cells from one body part to another because there is no way for them to be transported through circulation. Similarly, our results are in line with results described in PRP safety profile studies [17,29] but in contrary with Campbell et al. [30], where they described, in a systematic review of overlapping meta-analyses, an increased risk of local adverse reactions after multiple PRP injections, which was not the case in our study where PRP was injected only once. As in the results found in BMAC group, in our opinion, the lack of adverse events is logical because of the autologous biological property of PRP.

Efficacy of BMAC therapy for the knee OA was investigated in few different studies, comparing both results between the pre-intervention period and after the observation period [31,32,33,34], or comparing this therapy with placebo or physical therapy [11,12,35]. Results from these studies are controversial; some investigators proved the efficacy of BMAC therapy, whereas others found no differences between this therapy and placebo. We found somewhat better clinical results of BMAC therapy after 12 months in comparison with both competing regenerative therapeutic options and with the pre-intervention period. Ziegler et al. [36] compared anabolic, anti-inflammatory, and proinflammatory growth factors, cytokines, and chemokines in BMAC, blood, LP-PRP, and LR-PRP and found significantly higher concentration of interleukin 1 receptor antagonist (IL-1Ra) as compared with LR-PRP and LP-PRP, and this may be one of explanations for clinically better results of BMAC. These findings encourage us to use BMAC as the routine therapeutic option in our institutions.

We used the proximal tibia as the location for bone marrow aspiration. Narbona-Carceles et al. [37] found that the TNC concentration was significantly higher in the iliac crest (10.05 Millions/mL) compared with the tibia (1.7 Millions/mL), but the immunophenotype pattern of MSCs was similar for both locations. As the phenotype and differentiation potential of cells aspirating from the proximal tibia are similar to those from the iliac crest, and aspiration from the proximal tibia is a relatively easy and safe alternative to that from the iliac crest, we believe that even with a smaller number of cells, we could provide clinical benefit.

Regarding PRP therapy efficacy, there were more studies in the last decade and in literature, and we can find a number of randomized controlled trials (RCT), systematic reviews, and meta-analyses. Although many conclusions from these analyses emphasize the necessity for standardization and unification of PRP therapy, most of them are favorable toward the efficacy of PRP in comparison with placebo, HA, corticosteroids, and non-steroidal anti-inflammatory drugs [17,30,38,39]. The main competitor of PRP in these studies is HA. Beside a huge variety of PRP formulations and different conclusions about its efficacy, the predominant opinion is that LP-PRP is more effective than HA after 12 months. Leukocytes in PRP have been proposed to cause an exaggerated inflammatory response as they stimulate the release of interleukin (IL)-1β, IL-6, IL-8, and tumor necrosis factor alpha (TNF-α). Furthermore, leukocytes have also been thought to stimulate the production of reactive oxygen species that can lead to further muscle damage and inflammation [40]. In a study recently published by Ziegler et al. [36], it was found that LR-PRP had a significantly higher concentration of IL-1Ra than LP-PRP, and the conclusion was that in the cases where increased vascularity and healing were desired for pathological or injured tissues, including muscle and tendon, LR-PRP may be optimal, given its higher overall concentrations of PDGF, TGF-b, EGF, VEGF, and soluble CD40 ligand, but apparently not in the treatment of knee OA. In our study, we used LR-PRP as a therapy. Although, even in our study, PRP seems to be somewhat better than HA, there were no significant differences. As this is in contradiction with most of the previous studies, we believe that the main reason beyond this was a higher concentration of leukocytes in our preparations.

Our study had some limitations: First, it was not fully randomized study, as we explained in methodology section. Second, using LR-PRP, which in previous preclinical and clinical studies has been found as less effective in comparison with LP-PRP, was one of them. Heterogeneity of cellular preparations is broadly discussed in many studies, although we were striving to count cells and clarify cell properties as much as possible. Third, in the KOOS pain scale and IKDC, the BMAC group was different in comparison with the HA group before intervention, with less baseline pain in BMAC group, which could question if change in the scores is relevant, but we believe that results found in other scores with such a predominant superiority of the BMAC therapy could cancel these doubts. Moreover, even in KL grade, there were no differences in mean values among groups; in the BMAC group, we had less KL grade IV patients compared to those in the other two groups. We believe that, even with these limitations, BMAC shows somewhat better clinical improvement than the other two competitors.

## 5. Conclusions

Bone marrow aspirate concentrate, Leucocyte-Platelet Rich Plasma, and Hyaluronic acid injections are safe therapeutic options for knee OA and provide positive clinical outcomes after 12 months in comparison with findings preceding intervention. BMAC could be better in terms of clinical improvements in the treatment of knee OA than LR-PRP and HA up to 12 months. LR-PRP provides better outcomes than HA during the observation period, but these results are not statistically significant. More randomized controlled trials and high quality comparative studies are needed for direct correlative conclusions. 

## Figures and Tables

**Figure 1 medicina-57-01193-f001:**
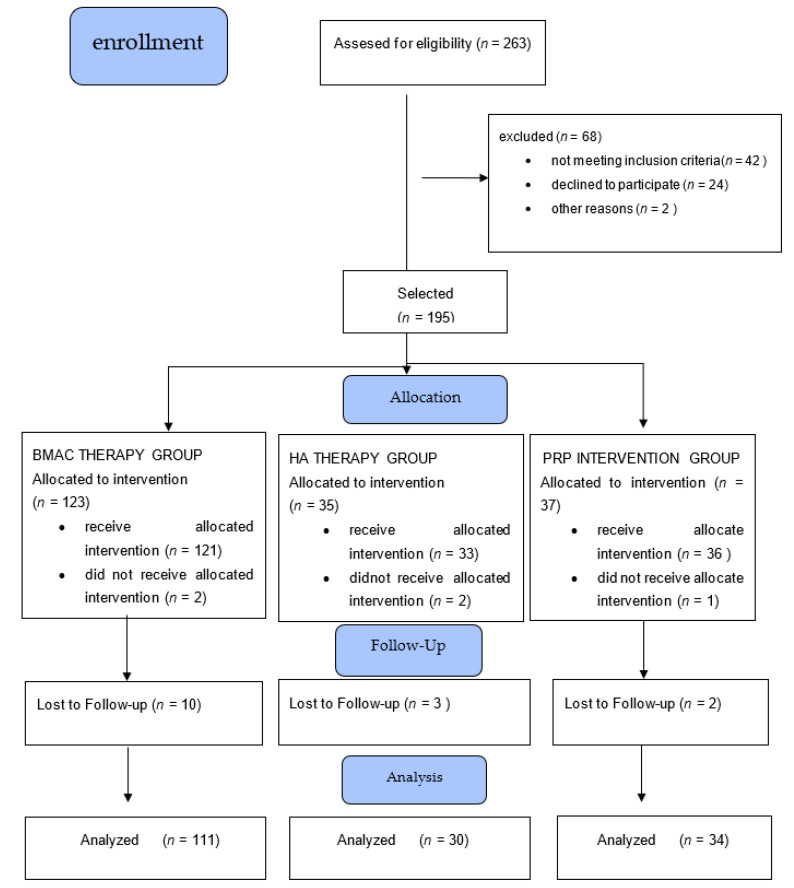
Flow diagram of the study.

**Figure 2 medicina-57-01193-f002:**
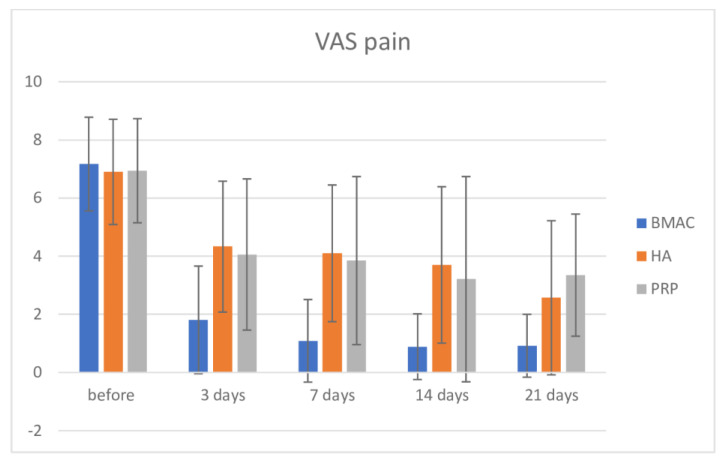
Visual Analog Scale pain before intervention and after 3, 7, 14, and 21 days.

**Figure 3 medicina-57-01193-f003:**
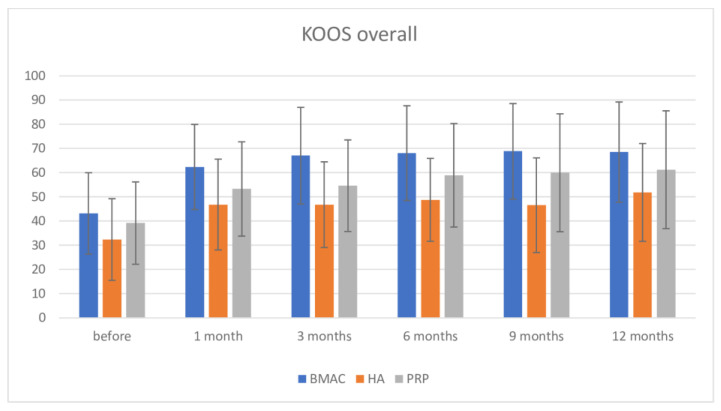
KOOS overall scale before intervention and after 1, 3, 6, 9, and 12 months.

**Figure 4 medicina-57-01193-f004:**
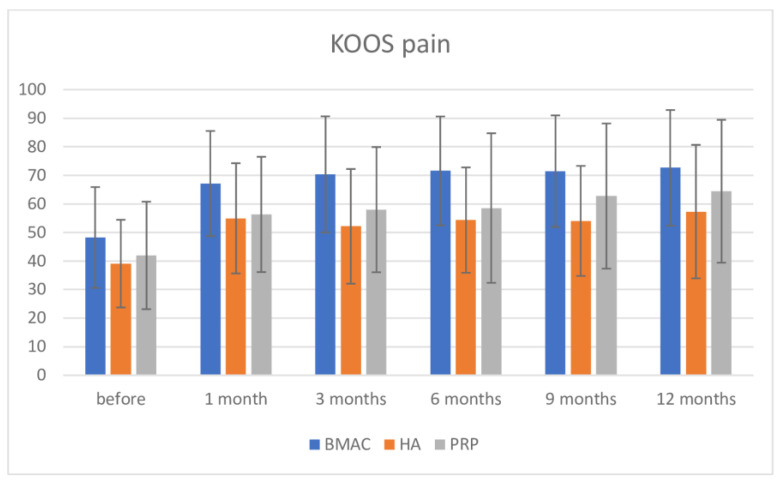
KOOS pain scale before intervention and after 1, 3, 6, 9, and 12 months.

**Figure 5 medicina-57-01193-f005:**
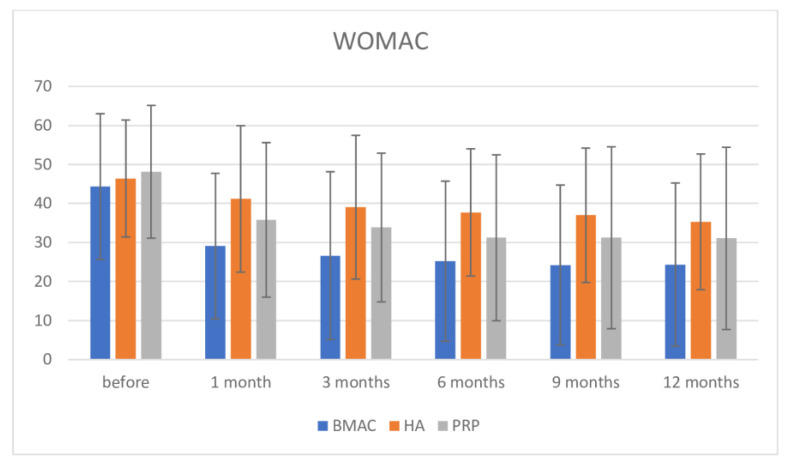
WOMAC scale before intervention and after 1, 3, 6, 9, and 12 months.

**Figure 6 medicina-57-01193-f006:**
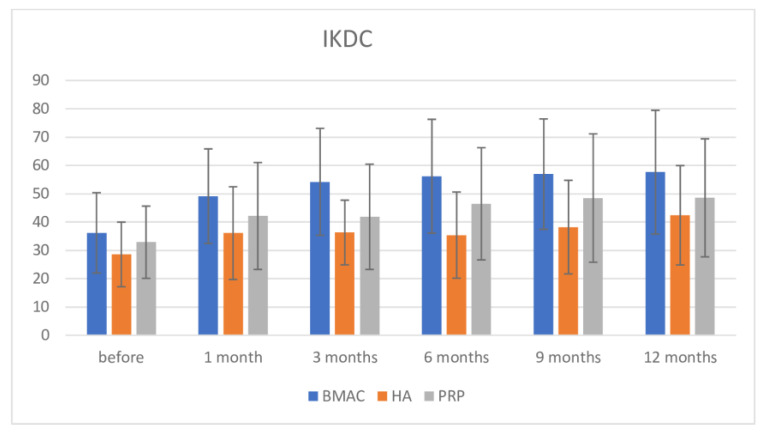
IKDC scale before intervention and after 1, 3, 6, 9, and 12 months.

**Table 1 medicina-57-01193-t001:** Patients characteristics for all intervention groups.

	BMAC	HA	PRP	*p*
Total Number	111	30	34	
Age, years (means ± SD)	56.9 ± 10.8	59.4 ± 14.0	58.8 ± 11.2	One way ANOVA F = 0.728; *p* = 0.485
Gender	Male: 57	Male: 13	Male: 15	Pearson Chi-Square = 0.943; *p* = 0.624
	Female: 54	Female: 17	Female: 19	
KL grade (%)	Grade 2: 49 (44.1%)	Grade 2: 13 (43.3%)	Grade 2: 12 (35.3%)	Pearson Chi-Square = 6.661; *p* = 0.155
	Grade 3: 46 (41.4%)	Grade 3: 8 (26.7%)	Grade 3: 12 (35.3%)	
	Grade 4: 16 (14.4%)	Grade 4: 9 (30.0%)	Grade 4: 10 (29.4%)	
BMI (means ± SD)	28.61 ± 4.53	29.98 ± 5.24	28.47 ± 4.54	One way ANOVA F = 1.136; *p* = 0.323
VAS pain sting	3.52 ± 3.07	3.97 ± 2.30	4.47 ± 2.64	One way ANOVA F = 1.492; *p* = 0.228
VAS pain injection	3.59 ± 2.85	3.57 ± 2.21	4.56 ± 2.69	One way ANOVA F = 1.741; *p* = 0.178

KL grade: Kellgren-Lawrence grade; BMI: body mass index; VAS: visual analog scale.

**Table 2 medicina-57-01193-t002:** Values of knee VAS scores in groups over time.

Intervention	Before	3 Days	7 Days	14 Days	21 Days
BMAC	7.17 ± 1.61	1.81 ± 1.85	1.09 ± 1.42	0.89 ± 1.13	0.92 ± 1.08
HA	6.90 ± 1.81	4.33 ± 2.25	4.10 ± 2.35	3.70 ± 2.69	2.57 ± 2.65
PRP	6.94 ± 1.79	4.06 ± 2.60	3.85 ± 2.89	3.21 ± 3.53	3.35 ± 2.10

**Table 3 medicina-57-01193-t003:** Values of clinical scores in intervention groups over time.

		Before	1 Month	3 Months	6 Months	9 Months	12 Months
KOOS pain	BMAC	48.24	67.13	70.35	71.55	71.46	72.63
	SD	17.66	18.4	20.32	19.07	19.53	20.25
	HA	39.09	54.95	52.14	54.33	54.04	57.32
	SD	15.35	19.3	20.08	18.44	19.28	23.37
	PRP	41.96	56.31	57.99	58.54	62.75	64.43
	SD	18.82	20.19	21.92	26.19	25.43	25.02
KOOS overall	BMAC	43.18	62.38	67.03	68.09	68.84	68.52
	SD	16.8	17.57	19.98	19.57	19.72	20.72
	HA	32.4	46.86	46.8	48.79	46.58	51.86
	SD	16.86	18.75	17.69	17.13	19.54	20.21
	PRP	39.19	53.29	54.63	58.94	59.98	61.24
	SD	17.01	19.49	18.91	21.39	24.36	24.34
WOMAC	BMAC	44.34	29.09	26.64	25.21	24.24	24.34
	SD	18.67	18.63	21.49	20.51	20.48	20.91
	HA	46.41	41.17	39.05	37.71	36.98	35.29
	SD	14.98	18.77	18.41	16.31	17.22	17.39
	PRP	48.12	35.8	33.84	31.21	31.22	31.06
	SD	17.02	19.8	19.05	21.26	23.3	23.34
IKDC	BMAC	36.19	49.13	54.2	56.18	56.94	57.62
	SD	14.18	16.7	18.87	20.1	19.48	21.84
	HA	28.57	36.09	36.31	35.39	38.22	42.4
	SD	11.43	16.39	11.43	15.22	16.52	17.55
	PRP	32.89	42.14	41.85	46.45	48.48	48.55
	SD	12.75	18.88	18.6	19.81	22.67	20.83

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
