# Peer review of "Bone Marrow Aspirate Concentrate versus Platelet Rich Plasma or Hyaluronic Acid for the Treatment of Knee Osteoarthritis"

_medicina, 2021, doi:10.3390/medicina57111193_

Round 1
Reviewer 1 Report
The reviewer would like to thank the authors for their efforts in performing the study. During the reviewing process, some points have been raised that require correction or clarification.
TITLE
Bone Marrow Aspirate Concentrate provides better clinical out- 2
come than Platelet Rich Plasma or Hyaluronic acid for the treat- 3
ment of knee Osteoarthritis: Please change the title without concluding anything in it. Just write down the technique you are using without any conclusions. Better use (Bone Marrow Aspirate Concentrate versus Platelet Rich Plasma or Hyaluronic acid for the treatment of knee Osteoarthritis)
ABSTRACT
The abstract is full of abbreviations without the introduction of the full-terms. Please add in the full-terms first followed by the abbreviations.
The authors have used several abbreviations for PRP as L-PRP, LR-PRP. Would you please unify the term and use only one?!
INTRODUCTION
Line 59, BMAC: What does the abbreviation stand for?
Aim of the study
Line 88-89, The aim of this study was to evaluate clinical effects of BMAC, L-PRP and HA therapy on knee OA: What kind of outcome to be evaluated? Please add to the aim of the study.
Line 91-92, our hypothesis was that BMAC could be: Please add the hypothesis under a separate title named "Null Hypothesis".
The reviewer would like to ask the authors about the study design of this study? Please clarify?!
If the study is a randomized clinical trials, then:
Line 96, This was a single-center, prospective clinical trial with 3 study arms: Clinical Trials are prospective in origin. Please remove prospective clinical trial from the sentence. The authors have to mention that it was a randomized clinical trial.
The authors have to mention that the study was performed according to the Declaration of Helsinki for medical research involving human subjects.
The authors have to mention that the study was performed according to the CONSORT guidelines and please cite.
The authors have to mention also the randomization technique of the patients and the program used as well as the allocation concealment of the patients and blinding.
Line 100-105: Please add the inclusion and exclusion criteria as bullet or numbers to be more clear to the readership.
Line 101-102, knee OA KL grade: The authors should write in details what is the KL grade!
Line 106-107, 195 patients who were diagnosed with OA, were included in this study: The authors have mentioned in the abstract that 175 patients were included in the study and now mentioning that 195 patients were included. What is the correct number of the patients included?
Line 120, about 80 ml of autologous bone marrow was aspirated: That's too much bone marrow aspiration. According to the literature, the maximum possible amount is 30 ml bone marrow.
Line 114, BMAC processing procedure: Did the authors perform any treatment to the BMAC before injection into the patients' knee. Please clarify? Was the BMAC subjected to any manipulation?
The authors had performed some in-vitro experiments on the BMAC. Would you please put those experiments as separate one under the title "In-vitro Experiments".
Line 53, Le: What does it stand for?
Line 170, 12 months later: Twelve months later.
RESULTS
The flowchart shows that 195 patients were included. However, the authors have mentioned, they were 175 patients. Please correct and clarify through out the whole manuscript.
Table 1: It would be better if the authors would put the percentage for the gender also.
Figure 2. Visual Analog Scale pain before intervention and after 3, 7, 14 and 21 days: It's somehow misleading with the VAS as the patients will be suffering from pain from the process of obtaining the BMA. How would you say that they have more pain before the operation and the pain decreased after the operation. The patients will be in pain from the invasive surgery of BMA. Please specify that VAS is used here for the pain only in the knee.
The authors should present the data either as tables or figures not as tables and figures to the same results as in figure 3, 4, 5 and 6 and table 3.
Author Response
title is changed in accordance with reviewer's recommendation.
TITLE
Bone Marrow Aspirate Concentrate provides better clinical out- 2
come than Platelet Rich Plasma or Hyaluronic acid for the treat- 3
ment of knee Osteoarthritis: Please change the title without concluding anything in it. Just write down the technique you are using without any conclusions. Better use (Bone Marrow Aspirate Concentrate versus Platelet Rich Plasma or Hyaluronic acid for the treatment of knee Osteoarthritis)
title is changed in accordance with reviewer's recommendation.
ABSTRACT
The abstract is full of abbreviations without the introduction of the full-terms. Please add in the full-terms first followed by the abbreviations.
The authors have used several abbreviations for PRP as L-PRP, LR-PRP. Would you please unify the term and use only one?!
Abbreviation issue is solved
INTRODUCTION
Line 59, BMAC: What does the abbreviation stand for?
abbreviations are removed
Aim of the study
Line 88-89, The aim of this study was to evaluate clinical effects of BMAC, L-PRP and HA therapy on knee OA: What kind of outcome to be evaluated? Please add to the aim of the study.
replaced with "clinical results"
Line 91-92, our hypothesis was that BMAC could be: Please add the hypothesis under a separate title named "Null Hypothesis".
it's done
The reviewer would like to ask the authors about the study design of this study? Please clarify?!
This study doesn't fullfil criteria for randomized trial and was designated as prospective study
If the study is a randomized clinical trials, then:
Line 96, This was a single-center, prospective clinical trial with 3 study arms: Clinical Trials are prospective in origin. Please remove prospective clinical trial from the sentence. The authors have to mention that it was a randomized clinical trial.
The authors have to mention that the study was performed according to the Declaration of Helsinki for medical research involving human subjects.
this is put in the manuscript
The authors have to mention that the study was performed according to the CONSORT guidelines and please cite.
As it's not designated as RCT, it's not neccesary to mention CONSORT guidelines. Tis study is performed in accordance with TREND criteria, I could mention or even upload TREND statement in supplamentary documents.
The authors have to mention also the randomization technique of the patients and the program used as well as the allocation concealment of the patients and blinding.
We put randomization technique for the part of the study where we randomized patients
Line 100-105: Please add the inclusion and exclusion criteria as bullet or numbers to be more clear to the readership.
Numbers are put
Line 101-102, knee OA KL grade: The authors should write in details what is the KL grade!
Kellgren-Lawrence scale. It's put now in text
Line 106-107, 195 patients who were diagnosed with OA, were included in this study: The authors have mentioned in the abstract that 175 patients were included in the study and now mentioning that 195 patients were included. What is the correct number of the patients included?
Overall number of patients selected in this study was 195, but during a process of intervention and follow up we lost 20 patients and final number of patientes trated and followed-up for 12 months is 175. This is also depcted in study flow chart, Figure 1.
Line 120, about 80 ml of autologous bone marrow was aspirated: That's too much bone marrow aspiration. According to the literature, the maximum possible amount is 30 ml bone marrow.
As we explained in methods, and in accordance both with our protocol and Arthrex Angel centrifuge manual, maximum volume of bone marrow for regular processing is 150 ml. We took 100 ml regularly, for geting about 6 ml of cell concentrate.
Line 114, BMAC processing procedure: Did the authors perform any treatment to the BMAC before injection into the patients' knee. Please clarify? Was the BMAC subjected to any manipulation?
No, BMAC was injected immediatly after centrifugation, in accordance with BMAC protocol without any other manipulation. This is regular Arthrex Angel procedure
The authors had performed some in-vitro experiments on the BMAC. Would you please put those experiments as separate one under the title "In-vitro Experiments".
It was put in the text
Line 53, Le: What does it stand for?
Leukocytes.. Now it is explained after the first appearance
Line 170, 12 months later: Twelve months later.
resolved
RESULTS
The flowchart shows that 195 patients were included. However, the authors have mentioned, they were 175 patients. Please correct and clarify through out the whole manuscript.
it was explained in Result chapter and Flow diagram. Now I put it also in Methodology
Table 1: It would be better if the authors would put the percentage for the gender also.
I will put it during a techinal finalization of tables
Figure 2. Visual Analog Scale pain before intervention and after 3, 7, 14 and 21 days: It's somehow misleading with the VAS as the patients will be suffering from pain from the process of obtaining the BMA. How would you say that they have more pain before the operation and the pain decreased after the operation. The patients will be in pain from the invasive surgery of BMA. Please specify that VAS is used here for the pain only in the knee.
It's put as "knee VAS pain scale"
The authors should present the data either as tables or figures not as tables and figures to the same results as in figure 3, 4, 5 and 6 and table 3.
I fully agree. Please note that In table 3., we put clear values. Figures were showed only for visualization of results ( and trends) without exact numbers. We can redact it but in this case some of useful data for overall understanding of clinical results could be missing.
Reviewer 2 Report
Interesting and relevant topic of study. Very thorough introduction.
- The fact that the study was not truly randomized should be addressed in the limitations. Additionally, any significant differences in the metrics at baseline for the BMAC group being less severe should be addressed since they had fewer KL 4 patients. Is there any potential bias from the first phase of patients knowing they were getting BMAC treatment, compared to the second phase when patients were randomized to HA or PRP?
- How many HA injections were received? 3 times every week for how many weeks and a total of how many treatments? If the BMAC and PRP groups only received 1 treatment, how do the VAS pain at 3, 7, and 14 days compare to when treatment occurred? Was the HA group still receiving injections during those reporting times or was that after the last injection in the HA group?
- It would be very helpful to denote significant differences on the figures with an asterisk to match the text.
- The results paragraphs are very difficult to follow with the interchanging of decimals and commas. In most cases, the commas should be decimals. Please change. What is MD in results?
- Difficult to read table 3 without any breaks between groups.
- Why do you think that the BMC group showed improvement in pain as early as day 3? Typically the BMAC should still be causing an inflammatory response at that point. The Leukocyte-rich PRP would elicit a strong inflammatory response at those early timepoints as well.
- Table 1: please define "VAS pain sting"
- What post-treatment rehab protocols were given or encouraged?
- Since the BMAC groups was significantly different than the other groups at baseline on some measures, you could consider reporting the delta score results.
Author Response
- The fact that the study was not truly randomized should be addressed in the limitations.
- It is addressed now in part of Discussion where we underline study limitations
- Additionally, any significant differences in the metrics at baseline for the BMAC group being less severe should be addressed since they had fewer KL 4 patients.
- It's addressed in same section with appropriate comment
- Is there any potential bias from the first phase of patients knowing they were getting BMAC treatment, compared to the second phase when patients were randomized to HA or PRP?
- We beleive no, because completely different people were treated without knowing each other or communicating about the study
- How many HA injections were received? 3 times every week for how many weeks and a total of how many treatments? If the BMAC and PRP groups only received 1 treatment, how do the VAS pain at 3, 7, and 14 days compare to when treatment occurred? Was the HA group still receiving injections during those reporting times or was that after the last injection in the HA group?
- for HA group, VAS scale was measured after 3 weeks, meaning after receiving the last injection which is recommended by HA producer as appropriate dosage for full clinical effects. We will put it in the text as clarification
- It would be very helpful to denote significant differences on the figures with an asterisk to match the text.
- Will be done during a technical revision on tables and figures
- The results paragraphs are very difficult to follow with the interchanging of decimals and commas. In most cases, the commas should be decimals. Please change.
- Also will be changed during a technical revision
- What is MD in results?
- Mean Differences. We will put it in results chapter prior the first appeareance
- Difficult to read table 3 without any breaks between groups.
- We will resolve it during a techical review of tables
- Why do you think that the BMC group showed improvement in pain as early as day 3? Typically the BMAC should still be causing an inflammatory response at that point. The Leukocyte-rich PRP would elicit a strong inflammatory response at those early timepoints as well.
- These results are something that we received from our patients not only during this study but also during a regular clinical work after the study. We beleive that BMAC provide very strong analgetic effect immediatelly after admission because of cellular protein properties that are excreted in the inflammatory enviroment after injection, as we discussed.
- Table 1: please define "VAS pain sting"
- We measured patients response to injection alone, it is not knee pain but injection pain and as we didn't found any clinical importance we didn't discuss it in details.
- What post-treatment rehab protocols were given or encouraged?
- it was regular protocol for knee osteoarthritis.
- Since the BMAC groups was significantly different than the other groups at baseline on some measures, you could consider reporting the delta score results.
- in accordance with our statistitician explanation, these distorstions in pre-intervention values were calculated in analyses of variances. If you insist, we could underline it but we beleive that our calculations together with discussion of this study limitation could be sufficient.
Round 2
Reviewer 1 Report
The Reviewer would like to thank the authors for their efforts in modifying the manuscript.
Author Response
I want to thank the Reviewer's efforts to provide me very usefull suggestions and remarks.
Reviewer 2 Report
I believe that readers will be interested in this work and it is an important contribution to the field. I believe there are some minor edits that will improve the quality of this manuscript.
This has been mentioned previously, but not completely fixed: please change references to L-PRP to LR-PRP, as you noted on line 163 (in methods and line 404), which is the industry standard.
Spell out Abbreviation SSD line 400.
Line 172-please change to a total of 3 treatments, 1 every week for 3 weeks.
VAS pain sting and VAS pain injection in Table. Please define/describe this metric at least in the footnote of the table if you do not want to discuss in the text. These are not intuitive.
Lines 351-353-It is a bit of an overstatement to say that because these were autologous treatments, " there is not one substance that could elicit foreign body reactions." First there is always the possibility of contamination. But second, there is documented evidence in the literature, although very rare, of a patients having an adverse reaction to their own autologous blood products. See Dome, R.E. Adverse reactions associated with autologous blood transfusion: evaluation and incidence at a large academic hospital. Transfusion 1998;38:301-306.
Author Response
I believe that readers will be interested in this work and it is an important contribution to the field. I believe there are some minor edits that will improve the quality of this manuscript.
Thank you for this observation and your's efforts to make our manuscript better
This has been mentioned previously, but not completely fixed: please change references to L-PRP to LR-PRP, as you noted on line 163 (in methods and line 404), which is the industry standard.
We removed all sufficient L and LR abbreviations, in all manuscript we call this therapeutic optin PRP but in Methodology section and Discussion we classify our foemulation in LR-PRP in accordance with ratio of cells, as it was proposend by Ehrenfest.Also, we put LR-PRP instead of L-PRP in all lines where L-PRP were mentioned.
Spell out Abbreviation SSD line 400.
It's done.
Line 172-please change to a total of 3 treatments, 1 every week for 3 weeks.
It's done in the sam line
VAS pain sting and VAS pain injection in Table. Please define/describe this metric at least in the footnote of the table if you do not want to discuss in the text. These are not intuitive.
That's right, it is now explained in Methodology section , line 177
Lines 351-353-It is a bit of an overstatement to say that because these were autologous treatments, " there is not one substance that could elicit foreign body reactions." First there is always the possibility of contamination. But second, there is documented evidence in the literature, although very rare, of a patients having an adverse reaction to their own autologous blood products. See Dome, R.E. Adverse reactions associated with autologous blood transfusion: evaluation and incidence at a large academic hospital. Transfusion 1998;38:301-306.
Correct, the statement "beside very rare cases" is put on the end of this statement together with proposed citation.